# Cryptoblend: An AI-Powered Tool for Aggregation and Summarization of Cryptocurrency News

**Andrea Pozzi** * , **Enrico Barbierato** and **Daniele Toti**

Department of Mathematics and Physics, Catholic University of the Sacred Heart, Via della Garzetta 48, 25133 Brescia, Italy
* Correspondence: andrea.pozzi@unicatt.it

**Abstract:** In the last decade, the techniques of news aggregation and summarization have been increasingly gaining relevance for providing users on the web with condensed and unbiased information. Indeed, the recent development of successful machine learning algorithms, such as those based on the transformers architecture, have made it possible to create effective tools for capturing and elaborating news from the Internet. In this regard, this work proposes, for the first time in the literature to the best of the authors' knowledge, a methodology for the application of such techniques in news related to cryptocurrencies and the blockchain, whose quick reading can be deemed as extremely useful to operators in the financial sector. Specifically, cutting-edge solutions in the field of natural language processing were employed to cluster news by topic and summarize the corresponding articles published by different newspapers. The results achieved on 22,282 news articles show the effectiveness of the proposed methodology in most of the cases, with 86.8% of the examined summaries being considered as coherent and 95.7% of the corresponding articles correctly aggregated. This methodology was implemented in a freely accessible web application.

**Keywords:** natural language processing; hierarchical clustering; text summarization; web development; noSQL database; blockchain; artificial intelligence; machine learning





## 1. Introduction

In the modern age of digital information, a vast amount of news articles are published every day by several online platforms, thus allowing worldwide users to easily access the latest news and be constantly updated even on highly specific issues. However, the immeasurable quantity of news, many of which is redundant, greatly limits the effectiveness of the efforts made to keep up to date with current events [1]. This issue is particularly accentuated for all those niches of people who rely on news as a work tool: politicians, financial market operators, diplomats, etc. In such sectors, the condensation and abstraction of information are increasingly needed. In particular, the goal of news summarization is to provide the users with only the essential information from the news in order to minimize their reading time: a shorter version of the different textual documents is thus created with the aim of preserving their fundamental contents [2].

In addition to the need to retrieve a summary of the news that is able to grasp the most salient aspects, another particularly important issue is the suitable selection of the news to read in terms of both the relevance of the topics that they address and their source. In fact, despite the enormous amount of news available online, many readers only consult a small subset of sources [3], mainly due to language barriers or simply habit. This results in a narrow perspective with a biased or incomplete perception of the information [4]. As a possible solution, automatic aggregators have been exploited in order to group similar news published by different newspapers, thus providing the reader with a broader spectrum of news items relating to a particular topic that, if considered in their entirety, can be free of bias.

As a consequence, it could be argued that the combination of aggregation and summarization techniques could prove beneficial to solving the problems mentioned above; for instance, both natural language processing (NLP) and machine learning methods are applied to identify groups of similar articles, from which, only the main concepts were extracted and proposed to the readers.

In this paper, for the first time in the literature to the best of the authors' knowledge, an automatic tool combining aggregation and summarization is presented for the case of financial news in the cryptocurrency field. In particular, state-of-the-art NLP solutions relying on the well-known transformers architecture [5] were implemented to solve both the aggregation and the extractive summarization phase, which were performed through the agglomerative clustering and the maximum marginal relevance approach, respectively. Moreover, a validation procedure was carried out by relying on expert readers to assess the performance of the proposed algorithm. Finally, a web application was developed to make the resulting condensed and unbiased information available to crypto-enthusiastic web users. The proposed web application is freely available at the following link: crypto-blend.xyz (accessed on 3 January 2023).

The rest of the paper is organized as follows. In Section 2 a literature review is presented to highlight the novelty of the proposed technique, whose methodology is described in detail in Section 3. The main results and the validation procedure are commented on in Section 4, while Section 5 concludes the paper.

## 2. Related Work

Since the debut of Bitcoin in 2009, the cryptocurrency industry has proven to be a rapidly growing industry. Portfolio diversification, extreme specialization, adaptability to different models, and privacy policies represent some of the strengths. However, Bitcoin is not free of risk: for instance, Barnes [6] reports specific issues mostly due to the lack of regulation, such as theft, initial coin offering (ICO) frauds (in the shape of pump and dump schemes), and the manipulation of market capitalization.

Cyber-attacks in the Bitcoin domain were discussed by Bratulescu et al. [7], who propose specific countermeasures achieved by monitoring financial transactions. Sureshbhai et al. [8] investigated a different approach articulated on a blockchain-based sentiment analysis framework (KaRuNa). Particularly interesting is the exploitation of long short-term memory (LSTM) to classify the scores produced during the sentiment analysis phase of the cryptocurrencies.

The so-called "bubbles" (i.e., periods when an anomalous behavior of markets is suddenly detected) can be used to investigate both investor and market characteristics. Sawhney et al. [9] propose CryptoBubbles, whose novelty consists of studying a crypto bubble multi-span prediction task and a text dataset of over 400 cryptocurrencies from seven exchanges distributed over five years. The dataset includes two million tweets. The crypto architecture is modeled by a multi-bubble hyperbolic network (MBHN) with different hyperbolic baselines to leverage the Riemannian manifold.

The scientific literature offers a vast range of tools implementing the aggregation and summarization of news. General news aggregators denote a limited service as they essentially play a passive role, i.e., they simply collect and display news from a set of sources. Typical examples are Google News (accessed on 3 January 2023), Yahoo!News (accessed on 3 January 2023), and HeadlineSpot (accessed on 3 January 2023), although the latter is present in the USA only. A detailed overview is discussed by Wojcieszak et al. [10], specifically for what concerns the venues: news websites, social networks, search engines, webmail, and hyperlinks. On a different level, other news aggregators adopt a more dynamic approach by presenting content that is better suited to the users, typically by asking them to fill out a questionnaire. Liotsiou et al. present the Junk News Aggregator (JNA, see [11]), focusing on the aggregation of junk news within social media such as Facebook. The tool is composed of (i) an Explorer, a module to filter and sort news;

(ii) Daily Visual Grid, providing a visual overview, and (iii) Daily Top-10 List displaying very popular posts for a specific day.

The objective of summarizing news collected from several sources is more challenging as it involves NLP-oriented processes (a review of the main statistical approaches used for news summarization can be found in [12]). This task includes, but is not limited to, the generation of a (usually) massive dataset accessed by a search engine.

For example, multi-news [13] is a large-scale news dataset based on a hierarchical model for neural abstractive multi-document summarization. It is produced by means of a pointer-generator network and a maximal marginal relevance (MMR) component used to evaluate sentence ranking scores according to relevancy and redundancy.

Varab et al. [14] reviewed different approaches to this problem, proposing a novel methodology articulated in manual annotation (to determine the news platforms that are suitable for a specific language), automatic collection, and quality control to filter out improper articles.

Kryściński et al. [15] proposed, in their article, a tool called CRTLsum, which is able to control the summarization of a text by allowing a user to specify a set of tokens. The tool architecture is based on pre-trained bidirectional and auto-regressive transformers (BARTs) [16], upon which the model is trained to learn the conditional distribution $p(y \mid x)$ ($x$ and $y$ denote the original document and its summary). As the output summary is built according to the control tokens $z$, the model aims at predicting the conditional distribution $p(y \mid x, z)$. The authors requested a few annotators to validate the output of CTRLsum across three different news datasets, reporting interesting accuracy levels concerning BART.

The authors in Gupta et al. [17] discussed a selection of pre-trained models based on the transformer architecture for text summarization. The analysis and comparison exploit a BBC news dataset. The models were evaluated according to a summary evaluation tool called Recall-Oriented Understudy for Gisting Evaluation (ROUGE) scores.

News Aggregator [18] is a tool used for collecting and summarizing news from reliable sources. After an initial aggregation, a pre-processing phase follows. The TextRank algorithm [19] is used to summarize the articles, resulting in a similarity matrix, which, in turn, is converted into a graph. This approach allows the procedure to classify the importance of a sentence by the number of edges. Finally, the ranks are sorted out in descending order and the results are checked out against ROUGE.

More specific tools implementing news aggregations and summarization in the cryptocurrency domain either produce or exploit sentiment analysis and relevance techniques to, for example, predict the price. Gadi et al. [20] introduced CryptoGDelt2022, a dataset generated by extracting news event from the Global Database of Events, Language and Tone (GDELT), including 243,422 rows and 19 columns. The cryptocurrency corpus was used to extract valuable statistics with regard to relevance, sentiments, and strength (i.e., the impact that news may have).

Rognone et al. investigated the impact of sentiment analysis on Bitcoin [21]. The data were sampled every 15 min by using Ravenpack News Analytics 4.0; in this way, the authors generated a sentiment index for each currency and Bitcoin. The currency returns, volume, and volatility were studied to see how they can be affected by the news sentiment. This goal was achieved by using an exogenous vector autoRrgressive model (VAR-X). Profiling cryptocurrency influencers is a task pursued by PAN (a series of scientific events and shared tasks on digital text forensics and stylometry): in https://pan.webis.de/clef23/pan23-web/author-profiling.html (accessed on 3 January 2023), the authors aim at categorizing aspects related to influencers, by using a low-resource setting, according to three subtasks: (i) low-resource influencer profiling, (ii) low-resource influencer interest identification, and (iii) low-resource influencer intent identification.

To evaluate the performance of text summarization tools, it is necessary to share a proper set of metrics. Fabbri et al. [22] re-visited some of the known measures, such as ROUGE, ROUGE-WE, $S^3$, BertScore, MoverScore, and many others. The authors also discussed a selection of models, regrouping them according to extractive and abstractive

levels. The results of this work were debated according to human evaluation across four dimensions (coherence, consistency, fluency, and relevance).

Table 1 summarizes the key aspects of the most interesting summarization tools.

**Table 1.** A comparative summarization tools summary.

| Summarization Tools | | |
| --- | --- | --- |
| **Tool Name** | **Architecture** | **Data Source Used** |
| Multi-news [13] | Based on a pointer-generator network and a maximal marginal relevance (MMR) component used to evaluate sentence ranking scores according to relevancy and redundancy | Newser (accessed on 3 January 2023) |
| CTRLsum [15] | Based on a neural network trained to learn the conditional distribution $p(y\|x)$, where $x$ and $y$ represent the source document and summary, respectively. | CNN/Dailymail news articles, arXiv scientific papers, and BIGPATENT patent documents |
| Based on a transformer paradigm [17] | Based on deep learning formalisms, such as recurrent neural networks (RNNs) | BBC news |
| News Aggregator [18] | Based on the Rank algorithm | BBC, World Health Organization |
| CryptoGDelt2022 [20] | Based on naive Bayes classification. The dataset is enriched with supervised machine learning scores for relevance, sentiment, and strength | Global Database of Events, Language and Tone (GDELT) |
| A generic tool [21] | Based on Ravenpack News Analytics 4.0 and an exogenous vector autoregressive model (VAR-X) | A leading global news database called RavenPack News Analytics, affiliated with Dow Jones Newswires. It analyzes relevant information from Dow Jones Newswires. |

Specifically, the main contribution of this manuscript is to combine cutting-edge natural language processing techniques, which are already available in the literature, to build, for the first time according to the knowledge of the authors, a general framework for the aggregation and summarization of news related to cryptocurrencies. The developed application aims to be the starting point for an integrated crypto news processing tool that can be extended in the future with several different features, such as a framework for correlating cryptocurrency prices to news sentiment reported by major online news outlets. With regard to [20,21], Cryptoblend explicitly focuses on the aggregation and summarization of financial news in the cryptocurrency domain.

## 3. Materials and Methods

In this section, the key points of the proposed methodology, conceptually schematized in Figure 1, are introduced. Specifically, Section 3.1 describes the process of continuously collecting news data related to cryptocurrencies from several online newspapers and storing them in an NoSQL database. In Section 3.2, the pipeline that is responsible for the elaboration of the collected data is presented. This pipeline consists of the following two steps: (i) an aggregation of similar news through a hierarchical clustering approach; and

(ii) an extractive summarization of the resulting aggregated text. Finally, the main technical details are reported in Section 3.3, which also discusses the implementation of a suitable web application able to make the summarized news (also called blends in the remainder of the discussion) available to the crypto community.

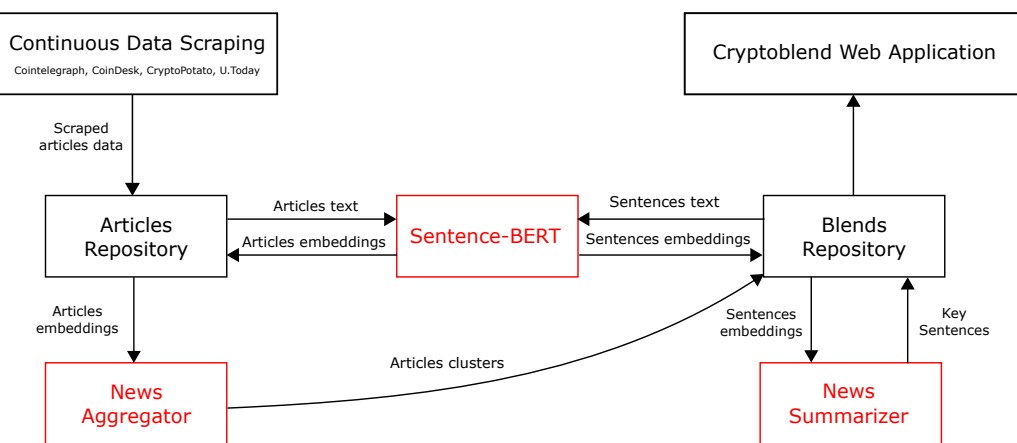

**Figure 1.** Conceptual scheme of the proposed methodology. Note that the module depicted in the upper-left corner (Continuous Data Scraping) implements the methodology described in Section 3.1, while the blocks highlighted in red correspond to those detailed in Section 3.2. Finally, the module in the upper-right corner (Cryptoblend Web Application) refers to the web application presented in Section 3.3.

### 3.1. Continuous Data Scraping

The news data are continuously collected through a web scraping process of $n_s$ online newspapers and the extracted information is stored in an NoSQL database (*news_db*). Specifically, for every newspaper, the scraping procedure is managed by a dedicated and properly implemented object that works as follows: (i) with a certain frequency ($f_s$), a request is made to the main page of the website in order to retrieve the URLs of recently published news; (ii) the URLs that are not already present in the database are then scraped in order to store data related to the title, text, and author in appropriate fields of the database entries. Moreover, the picture related to each article, if available, is also downloaded and stored in a local directory of the server (note that a unique identifier, *picture_id*, is assigned to each scraped picture and inserted in the database in order to recover the picture later on).

The scrapers for the different newspapers are executed in a multi-threaded fashion to minimize the scraping time, while suitable pauses are introduced to prevent the remote websites from being overwhelmed with requests. Each entry of the *news_db* consists of an Article object structured as follows:

$$Article : \{url : str;$$
$$title : str;$$
$$text : str;$$
$$author\_name : str;$$
$$author\_link : str;$$
$$picture\_id : int;$$
$$scraped\_ts : float\}$$

where *scraped_ts* represents the time stamp in which the considered article has been scraped.

### 3.2. Data Elaboration Pipeline

In parallel to the scraping process, a pipeline devoted to the elaboration of the data is executed. The results of such a pipeline are called blends, i.e., concise textual elements

that summarize all of the news, related to a particular topic, that has been published by the considered newspapers in a certain period. Specifically, a blend is created by aggregating the text of similar news and extracting the key sentences of the resulting text according to their relevance. Both the aggregation and the summarization procedures rely on a transformer-based framework able to retrieve state-of-the-art sentence, text, and image embeddings. Specifically, such embeddings are obtained through the well-known bidirectional encoder representations from transformers (BERTs) [23], which is a language representation model designed to retrieve deep bidirectional representations from the unlabeled text.

Moreover, it is important to also recall that the blends are stored in an NoSQL database (*blends_db*), in which each entry is a Blend object structured as follows:

$$
\begin{aligned}
Blend : \{ & title : str; \\
& articles : list[Article]; \\
& summary : str; \\
& blend\_embedding : list[float]; \\
& picture\_id : int \\
& last\_update : float; \\
& keywords : list[str] \}
\end{aligned}
$$

where the title of the blend corresponds to that of the first article in the articles list, which contains all of the Article objects constituting the blend, while the summary field represents the summarized text, obtained by extracting the key sentences from the original articles. The *blend_embedding* field provides a vectorized representation of the blend based on the average of the embeddings of the original articles. The *picture_id* is that of the first article in the blend with an available picture. If a blend ends up being composed of a set of articles without an image, a relevant picture is displayed, which is automatically selected according to the most frequent keywords in the blend. This is carried out because a potentially relevant image may contain in itself additional information that may prove useful to the users of the web application. Finally, the *last_update* field refers to the time stamp in which the blend is updated for the last time, while the keywords field consists of a list of the named entities extracted from the blend summary, where potential duplicates are discarded.

It is important to underline that the data elaboration pipeline is repeatedly executed every $f_p$ seconds.

### 3.2.1. News Aggregation

The purpose of this pipeline step is to understand which articles deal with the same topic. In order to achieve this goal, it is essential to establish a time window within which similar articles can be aggregated. The set of recent articles $A(t)$ is defined as the articles that have been scraped in the last $t_{ra}$ days, and the set of recent blends $B(t)$ as the blends that have been created or updated in the last $t_{rb}$ days, i.e.,

$$
A(t) = \{ a \in news\_db : t - a['\text{scraped\_ts}'] < t_{ra} \cdot 24 \cdot 3600 \}, \tag{1a}
$$

$$
B(t) = \{ b \in blends\_db : t - b['\text{last\_update}'] < t_{rb} \cdot 24 \cdot 3600 \}, \tag{1b}
$$

where $t$ is the current time stamp.

Every article $a_i \in A(t)$ is then processed by a BERT-based module to retrieve its embedding $a_{d,i} \in \mathbb{R}^{n_d}$, where $n_d$ is the size of the embeddings. Hence, the embedding matrix for the recent articles is built as follows:

$$
E_a = [a_{d,1} \cdots a_{d,n_a}]^\top, \quad n_a = |A(t)|. \tag{2}
$$

Similarly, the matrix containing the embeddings of the recent blends can be built as follows:

$$E_b = [b_{d,1} \cdots b_{d,n_b}]^\top, \quad n_b = |B(t)|. \tag{3}$$

where $b_{d,i}$ is the embedding of the *i*-th blend, i.e., $b_i \in B(t)$. Note that, given a specific blend $b_i$, consisting of the articles $a_1, \ldots, a_{n_{b_i}}$, its embedding $b_{d,i} \in \mathbb{R}^{n_d}$ is computed as the average of the embeddings of the different articles in the blend, i.e., as follows:

$$b_{d,i} = \frac{1}{n_{b_i}} \sum_{j=1}^{n_{b_i}} a_{d,j} \tag{4}$$

where $a_{d,j} \in \mathbb{R}^{n_d}$ is the embedding of the *j*-th article in the blend.

Then, agglomerative clustering [24] is carried out on the matrix $E \in \mathbb{R}^{(n_a+n_b) \times n_d}$, which results from the concatenation of the articles and blends embedding matrices (i.e., $E = [E_a^\top, E_b^\top]^\top$), with the aim of grouping together the rows that have similar semantics, i.e., articles and blends that address the same topic. Such a clustering approach is a type of hierarchical clustering exploiting a bottom-up approach: each data point starts from its own cluster, and these clusters are then joined greedily by taking the two most similar ones together and merging them. A threshold on the cosine similarity between clusters is used to terminate the algorithm. Note that a proper choice of such a threshold is required to address the issue of potential redundancy of the blends. According to the clustering outcome, the existing blends are updated or, if needed, new ones are generated. In particular, the following scenarios may occur in a given cluster:

- Case 1: there are no blends in the cluster. A new blend is generated from the articles of the cluster;

- Case 2: only one blend is present in the cluster. If there are no articles in the cluster, the blend is left unchanged; otherwise, is updated by appending the articles in the cluster to the list contained in the articles field and modifying the *blend_embedding* one to be the average embedding of the contained articles;

- Case 3: two or more blends are present in the cluster. If there are no articles in the cluster, all of the blends are left unchanged; otherwise, only a randomly selected blend is updated as carried out for the Case 2, while the other blends are left unchanged;

### 3.2.2. News Summarization

In this section, an extractive summarization procedure is described in order to fill or update the summary field of the blends that have been created or updated in Section 3.2.1.

Specifically, for each blend, the texts of the corresponding articles are concatenated into a unique document (*D*), from which a limited number of key sentences are extracted by relying on a maximum marginal relevance approach [25]. This latter, which has been largely employed for text summarization, aims to reduce redundancy and increase diversity in the resulting summary, by selecting the most important sentences according to a combined criterion of relevance and novelty of information.

The summarization approach works as follows:

1. Invoke BERT to compute the embedding *D* of the whole document and the embedding $s_i$ of every sentence in the document;

2. Consider $\mathcal{R}$ as the set of all of the sentences in the document, $\mathcal{S}$ as the set of sentences selected for the summarization (initialized as an empty set), and $\mathcal{R} \setminus \mathcal{S}$ as the set of non-selected sentences in $\mathcal{R}$;

3. Rank the sentences in the document according to maximum marginal relevance by repeatedly solving the following optimization problem:

$$\max_{s_i \in \mathcal{R} \backslash \mathcal{S}} \left[ \lambda sim(s_i, D) - (1 - \lambda) \max_{s_j \in S} sim(s_i, s_j) \right] \tag{5}$$

where $\lambda \in \mathbb{R}$ is a tuning parameter and $sim(a, b)$ computes the cosine similarity between the documents $a$ and $b$;

4. Select the desired number of sentences with the highest rank to form the summary of the document.

The items in the set $\mathcal{S}$ of the extracted key sentences are finally joined to form a unique *str* object that constitutes the summary field of the blend.

### 3.3. Implementation of the Web Application

The main details related to the implementation of the scraping and data elaboration procedures are discussed here, together with a brief description of the web application pattern and tools.

#### 3.3.1. Technical Details for Scraping and Data Elaboration

Specific details on the implementation of the methodology are reported here in order to enhance the reproducibility of the results discussed. First of all, it is important to note that, in the proposed tool, the following $n_s = 4$ online sources are considered for the web scraping procedure:

- Cointelegraph (accessed on 3 January 2023);
- CoinDesk (accessed on 3 January 2023);
- U.Today (accessed on 3 January 2023);
- CryptoPotato (accessed on 3 January 2023).

and that such a procedure is performed by relying on the well-known Python library BeautifulSoup [26] with a scraping frequency of $f_s$ = 900 s. Note that the considered data sources are international newspapers that mainly publish articles related to cryptocurrencies and decentralized finance, with a monthly number of visits that ranges from 1 million (CryptoPotato) to 15 million (CoinDesk). Moreover, MongoDB [27] collections are exploited as NoSQL databases for storing the collected and processed data by relying on the Python library PyMongo and the cloud service MongoDB Atlas. Within this context, note that the Article and Blend objects are implemented as a Python dataclass.

The BERT-based module exploited in the data elaboration pipeline relies on Sentence-Transformer [28], a modification of the pre-trained BERT network that uses Siamese and triplet network structures to derive semantically meaningful sentence embeddings (with size $n_d = 768$) that can be compared using cosine-similarity. The aggregation procedure is based on the implementation of the agglomerative clustering proposed by the well-known Python module scikit-learn [29] (through *AgglomerativeClustering* class), considering the following parameters: $t_{ra}$ = 3 and $t_{rb}$ = 5. Moreover, it is important to highlight that the data elaboration pipeline undergoes iterations every $f_p$ =180 s. Finally, the tuning parameter in the maximum marginal relevance algorithm is chosen as $\lambda = 0.5$.

#### 3.3.2. Web Application

The web application discussed in this paper is structured according to the three-tier pattern (commonly used for e-commerce and business-to-consumer websites, and, more in general, for applications requiring an interaction layer with external users), where each of the following tiers is assigned a specific role:

- A presentation layer consisting of a front-end web server, serving static and cached dynamic content that are rendered by the browser. This level relies on documents written in the well-known HyperText Markup Language (HTML), assisted by Cascading Style Sheets (CSS) [30] to define the content style and layout, as well as JavaScript [31] to make the pages interactive. Note that the pages' style and layout are based on the CSS template provided by Bootstrap [32], which is an open-source front-end frame-

work for faster and easier web development. In particular, Bootstrap includes HTML and CSS-based design templates for typography, forms, buttons, tables, navigation, modals, image carousels, and many others;

- A middle logic layer for dynamic content processing and generation based on FastAPI (accessed on 3 January 2023), which is a modern high-performance web framework for building APIs based on standard Python type hints;
- A data layer, made up of a back-end database and Create/Retrieve/Update/Delete (CRUD) APIs, including both data sets and the database management system software that manages and provides access to the data. This tier is based on MongoDB Atlas, which is a global cloud database service built and run by MongoDB Inc., New York, NY, USA.

### 3.3.3. Availability

The web application is publicly available and freely accessible at the following URL: https://cryptoblend.xyz (accessed on 3 January 2023).

## 4. Results and Discussion

In this section, the main results achieved by developing the proposed tool are presented. Specifically, in Section 4.1, the features of the web application cryptoblend.xyz (accessed on 3 January 2023) are detailed, with a specific focus on the user experience and interface, while, in Section 4.2, relevant statistics related to the collected and processed information are reported. Finally, in Section 4.3, the proposed methodology is validated by domain experts. The results show that Cryptoblend may be deemed a reliable integrated framework for the natural language elaboration of cryptocurrencies news, able to collect the consensus of expert readers on the accuracy of both the news aggregation and summary phases.

### 4.1. User Experience and User Interface

The considered web application consists of two main pages: the home page (also called news feed in this context) and the blend page. These pages are described in details in the following.

The web application's homepage enables users to browse the extracted information by resorting to the so-called infinite scroll paradigm, implemented via a JavaScript plugin that automatically adds and presents each subsequent page, preventing the application from loading the whole set of contents all at once and thus decreasing loading times and memory usage.

From a business-oriented point of view, this choice is also based on the assumption that, by preventing the users from explicitly taking the action of loading a new page, the retention rate from the users themselves is increased. Aside from improving the user experience, the infinite scroll concept decreases both network and energy consumption.

A description of the home page of the presented website is provided in Figure 2, where the following elements can be noted:

- The website title *cryptoblend* in the header, which allows the user to refresh the page if clicked (1);
- A navigation bar to navigate between the news feed and the about page (2);
- A button that allows the user to sort the news according to their relevance (computed as a weighted score between number of articles and elapsed time from the last update) or according to their last update instant (3);
- The boxes of the different blends constituting the news feed that can be uploaded in groups of ten due to the exploitation of the infinite scroll paradigm (4).

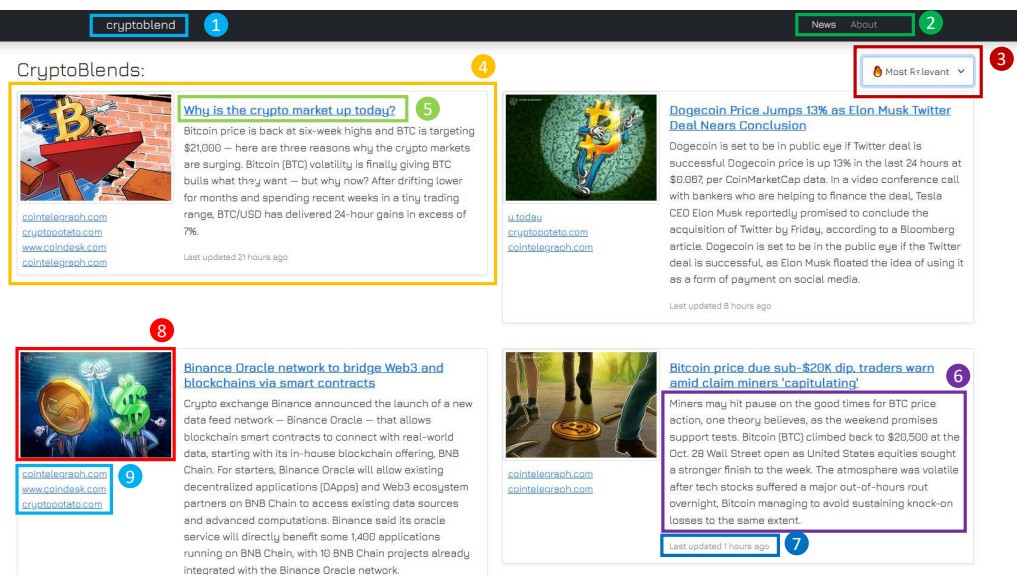

**Figure 2.** Graphics of the web application home page, in which the different parts can be described as follows: (1) website title, (2) navigation bar, (3) sorting of the news according to last update or relevance, (4) blend's box, (5) title of the blend, (6) blend's extractive summary, (7) time stamp of the blend's last update, (8) blend's image, and (9) list of the original articles.

Moreover, each blend's box contains:

- The title of the blend (5);
- The text corresponding to the extractive summary of the original articles (6);
- The time of the last update for the blend (7);
- A picture describing the content addressed by the text, which is taken from one of the articles that constitute the blend (8);
- A list of the url addresses of the original articles (9).

Note that each blend can be opened in a separate page (shown in Figure 3) by clicking on its title. In the blend page, all of the elements that are inside the corresponding blend box in the home page are reported. Moreover, a button enabling content sharing through social media (Facebook and Telegram) is also made available (10).

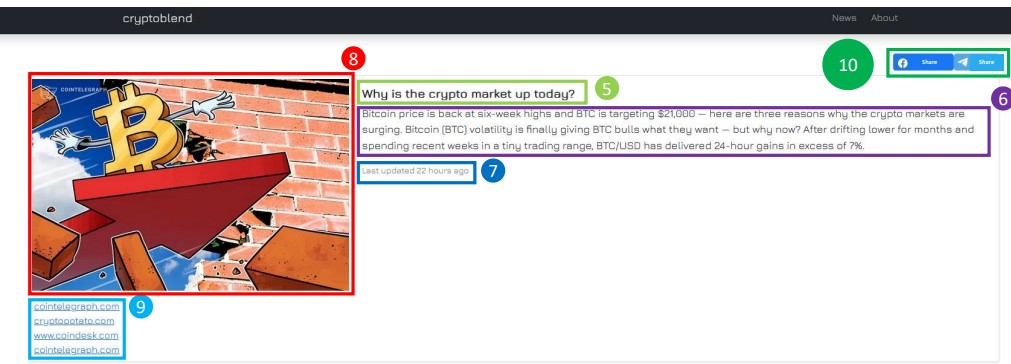

**Figure 3.** Graphics of a blend's page, in which the different parts are enumerated as in Figure 2, with the exception of the button (10) that allows for sharing the blend's content on the social media.

### 4.2. Statistical Analysis of Collected and Elaborated Data

The web application proposed in this paper has been continuously online since 3 March 2022 and has collected, aggregated, and summarized a total of 22,282 articles, resulting in 12,599 blends (as of 31 October 2022). The average daily scraping frequency for the different newspapers is as follows: 26.3 articles per day for *Cointelegraph* (with standard deviation std = 1.7), 25.6 articles per day for *CoinDesk* (std = 1.2), 11.4 articles per day

for *CryptoPotato* (std = 0.8), and 28.3 articles per day for *U.Today* (std = 3.1). As far as the blends statistics are concerned, it is interesting to note that the daily average number of created blends is 51.4 (std = 3.7), where the distribution of the number of original articles per blend is reported in Figure 4. Specifically, such a distribution appears to be skewed and heavy-tailed on the right, as expected from the fact that the number of articles per blend is bounded from below (a blend must contain at least one article by definition). Finally, effort has also been devoted to downloading and storing the pictures related to the scraped articles, when available, thus coupling the retrieved textual content with its visual description. In particular, more than 13,000 pictures have been collected during the activity period of the website, with an average daily frequency of 54.6 elements (std = 5.8).

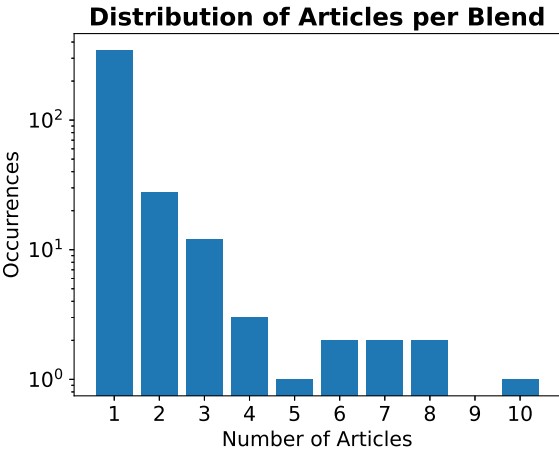

**Figure 4.** Distribution of the number of articles per blend in logarithmic scale.

In the following discussion, the focus is devoted to the textual and linguistic aspects of the data collected and processed. In detail, the distribution of the number of tokens obtained by applying the SpaCy [33] tokenization pipeline to both the articles and the blends is depicted in Figure 5, where a token in the considered corpus consists of 5.09 characters on average. Beyond the obvious result that the average number of tokens is much larger for the articles than for the blends, it is interesting to note that the number of tokens in the articles exhibits a right-skewed distribution, while its distribution for the blends presents a bell-shaped curve, thus highlighting the normalizing effect obtained through the proposed extractive summary.

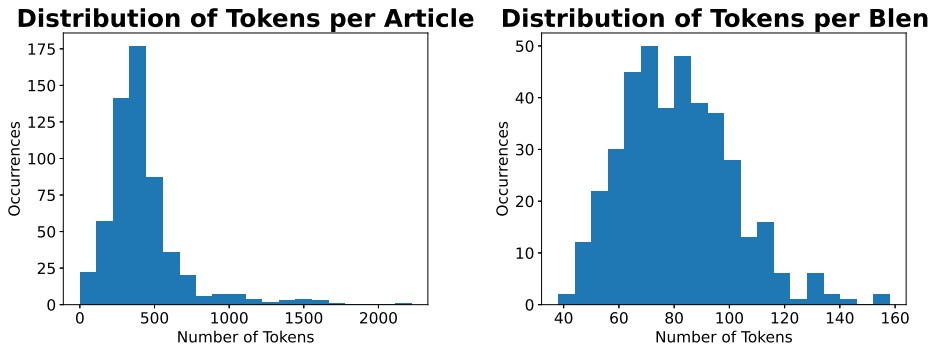

**Figure 5.** Distribution of the number of tokens in the articles (**left**) and in the blends (**right**). Specifically, the average number of tokens in an article is 418.46, with standard deviation 211.54, whereas, for a blend, the average number of tokens is 85.92, with standard deviation 21.63.

In Figure 6, the occurrences of the most common keywords associated with the blends generated in the first week of November 2022 are presented. In particular, the keywords are extracted through the named entity recognition pipeline of SpaCy. As expected, they mainly refer to important news circulated in those days about "FTX, a famous crypto

exchange, filing for bankruptcy due to lack of liquidity". The interested reader can find a blend that provides a summarized description of the facts at the following link: "Breaking: FTX Allegedly Stops Processing Withdrawals" (accessed on 3 January 2023).

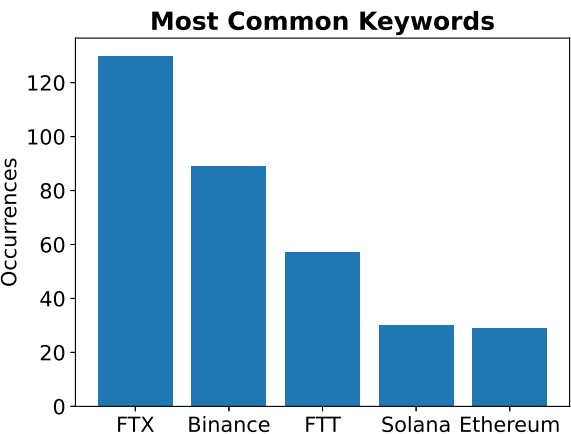

**Figure 6.** Most common named entities in the retrieved summaries, obtained through the named entity recognition pipeline of SpaCy.

*4.3. Validation of the Data Elaboration Pipeline*

In this subsection, empirical validation of the proposed methodology is carried out in order to highlight both its effectiveness and critical aspects. In particular, 200 randomly extracted blends were analyzed by two experienced readers who independently expressed an evaluation of the aggregation and summary capabilities of the tool presented in this paper. Such an analysis highlighted that the quality of the aggregation and summarization procedures is generally high. Nevertheless, situations have been encountered in which the algorithms can be further improved. In fact, the results show that 4.3% of the original articles were aggregated to a blend incorrectly, in the sense that the topic covered was not exactly in line with that of the blend. As far as the summarization algorithm is concerned, the percentage of the blend's summary that has been considered incomplete or incorrect by the experienced readers is 13.2%. Note that, although the readers were selected by the authors due to their knowledge in the field of decentralized finance and cryptocurrencies, such validation may produce biased results due to the subjective nature of human perception. A possible solution may be to rely on a number of human validators that is large enough to statistically reduce the aforementioned bias, or, alternatively, on objective metrics suitable for automatic validation.

In order to provide an example of a blend that has been positively evaluated, the one whose title is "Breaking: Largest Cryptocurrency Options Exchange Deribit Suffers $28 Million Hack" (accessed on 3 January 2023) is considered here, with a summary denoted as follows:

> "Deribit, the largest cryptocurrency options exchange, has suffered a $28 million hack, according to an announcement posted on Twitter. It is unclear when the exchange will be able to reopen. The exchange has stressed that it remains in a strong financial position, and the recent hack will not affect its operations."

obtained by four different original articles:

- U.Today Article (accessed on 3 January 2023): "Breaking: Largest Cryptocurrency Options Exchange Deribit Suffers $28 Million Hack";
- Cointelegraph Article (accessed on 3 January 2023): "Deribit crypto exchange halts withdrawals amid $28M hot wallet hack";
- CoinDesk Article (accessed on 3 January 2023): "Crypto Exchange Deribit Loses $28M in Hot Wallet Hack, Pauses Withdrawals";

- CryptoPotato Article (accessed on 3 January 2023): "Crypto's Biggest Options Exchange Deribit Hacked for $28 Million, Loss Covered by Company Reserves".

As can be noticed, all of the articles address the same topic, i.e., "the recent $28 million hacking suffered by Derebit, an important crypto exchange", thus confirming the effectiveness of the proposed agglomerative clustering in aggregating articles. Note that the title of the blend is chosen to be equal to the title of the first article in the blend's list of articles. As far as the summarization is concerned, an analysis of the most frequent terms used in the text of the articles and the blend itself is conducted here by relying on the concept of word clouds [34]. These provide a quick and visual overview of the main topics discussed in a body of text by graphically depicting the different words so that the font size of a word is positively correlated with its frequency. As can be noted from Figure 7, the word cloud obtained from the blend text (on the right) is similar to the one retrieved from the original articles; specifically, in both of them, "Derebit" is recognized as a large crypto exchange. However, the "hack" concept is not so prominent in the word cloud of the original articles, whereas it is correctly of primary importance in the blend's word cloud.

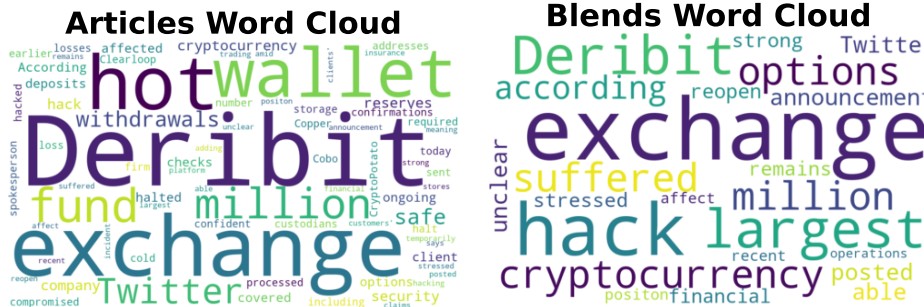

**Figure 7.** Comparison between the word clouds obtained by the original articles (**left**) and the resulting summary contained in the corresponding blend (**right**).

It is also interesting to show an example of a blend for which the aggregation process has been considered incorrect, as is the case, for instance, for the blend that can be found at the following link—Stablecoin Issuer Paxos Receives Operating License From Singapore Regulator (accessed on 3 January 2023)—which is based on the following articles:

- CoinDesk Article 1 (accessed on 3 January 2023): "Stablecoin Issuer Paxos Receives Operating License From Singapore Regulator";
- CoinDesk Article 2 (accessed on 3 January 2023): "Singapore Grants Stablecoin Issuer Circle In-Principle License to Offer Payment Products";
- Cointelegraph Article (accessed on 3 January 2023): "Stablecoin issuers Circle and Paxos gain approvals in Singapore";
- CryptoPotato Article (accessed on 3 January 2023): "Circle and Paxos Secure Regulatory Approval in Singapore".

Specifically, all of the articles refer to the general concept of "companies that obtained licenses from Singapore's financial regulator", but the first article is related only to a company named "Paxos", whereas the last three articles also deal with another company named "Circle". Although the number of erroneously aggregated articles seems negligible, countermeasures should be considered in the future to prevent this problem from affecting the consistency of the output of the summarization process. In particular, a possible solution may be the use of a rule-based filter that, among the articles that are candidates to be in a certain cluster, selects only those that have in common a number of keywords above a certain threshold, which should be properly tuned to maximize the overall performance.

Finally, an example of a summary that can be improved is the one related to the following blend—"Binance's CZ Confirms Participating as Equity Investor in Musk's Twitter Takeover" (accessed on 3 January 2023)—which results as follows:

> *"Binance confirmed on Friday that it was as an equity investor in billionaire technology entrepreneur Elon Musk's takeover of microblogging service Twitter (TWTR). CZ said in a tweet that Binance had wired some $500 million as part of the deal two days ago. The Binance CEO said the company wired $500 million to take a share of equity as Elon Musk's Twitter takeover is finally sealed."*

It is clear that the three sentences are actually the reformulation of the same concept, i.e., "Binance participated as an investor in Elon Musk's Twitter takeover with $500 million" (which, correctly, is the subject of all of the seven articles linked to the considered blend). The use of redundant sentences in the summary could be avoided by relying on a more appropriate tuning of the $\lambda$ parameter in the maximum marginal relevance algorithm; see Equation (5). Such a parameter is now only selected based on the human perception of experienced domain readers seeking to gauge the quality of the summary [35]. In this regard, further developments are planned to be devoted to the implementation of objective metrics to assess the performance of the summarization algorithm, with the aim of devising an automatic tuning mechanism of the $\lambda$ parameter.

## 5. Conclusions

In this paper, the development of a methodology able to aggregate and summarize news articles related to cryptocurrencies, the blockchain, and decentralized finance was proposed. This methodology was implemented in a fully functioning publicly available web application. The application exploits a BERT-based sentence representation, in combination with an agglomerative clustering algorithm, to identify groups of similar articles. In addition, a maximum marginal relevance approach was used to extract the most representative sentences of each cluster. The effectiveness of the presented methodology and application was highlighted through a validation procedure carried out by expert human readers against a set of randomly selected documents. The results show that 86.8% of the examined summaries were considered to be coherent and that 95.7% of the corresponding articles were correctly aggregated.

Extensions of this work may include an analysis aimed at identifying novel metrics, which could be suitable for the performance evaluation of unsupervised methods, such as the ones used in news clustering and extractive summarization. In this way, it could be possible to perform an automatic tuning of the hyper-parameters without relying on the potentially biased judgment of human experts.

**Author Contributions:** Conceptualization, A.P.; data curation, A.P.; investigation, A.P.; methodology, A.P.; software, A.P.; validation, A.P., E.B. and D.T.; writing—original draft, A.P., E.B. and D.T.; writing—review and editing, A.P., E.B. and D.T. All authors have read and agreed to the published version of the manuscript.

**Funding:** This research received no external funding.

**Institutional Review Board Statement:** Not applicable.

**Informed Consent Statement:** Not applicable.

**Data Availability Statement:** Data sharing not applicable.

**Acknowledgments:** Grateful acknowledgement is also made to Massimo Zambelli for his contribution in the development of the presented web application and for his valuable suggestions.

**Conflicts of Interest:** The authors declare no conflict of interest.

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
