# Peer review of "Cryptoblend: An AI-Powered Tool for Aggregation and Summarization of Cryptocurrency News"

_informatics, doi:10.3390/informatics10010005_

Round 1

Reviewer 1 Report

A well written and interesting manuscript. The overall presentation is easy to follow and understand. The language used is acceptable in general. However, the paper does not propose any new idea or methodology.  It builds a web application using existing components. Therefore, it is difficult to categorize it as a research article. If there are any new ideas or novelty, the authors must revise the manuscript to emphasize that aspect of the work.

The following are the recommendations to the authors to help improve the manuscript.

·       There are minor mistakes/problems in the language used.

o   Colloquial/awkward language used: Example: “few simple clicks”, “unnecessary waste of time”, “we are going to describe such pages in detail”, “thanks to the”’ “The tool hourly queries a list ..” etc

o   Some sentences are too long and difficult to understand. Example: “On the one hand, the situation ……with current events[1]”, “Extensions of this work …”, “More specific tools implementing …”

o   There are a few “small” mistakes such as  use of each instead of every (Example: The data is sampled each 15 minutes”) , into instead of in (Example: “resulting into”) and  in instead of as, redundant use of also, analisys etc.

·       Table 1  presents both corpora and tools but the caption reads “A comparative tool summary”, the short descriptions column does not provide “description (Example: “Simple task. No need to mee users expectations”) does not describe a tool. the descrion and the data source must be revised and if necessary additional references must be given.

·       Systems such as BERT and BART are used without giving the full form. The manuscript should be scanned for similar omissions related with abbreviations.

·       In the materials and methods section, the architecture of Cryptoblend and the data elaboration pipeline used must be presented in a diagrammatic form (block diagram) to provide the big picture. Each component must then be described. Specifically since the system involves a BERT-based module, BERT must be presented to the readers as part of this section. How the “BERT-based module”  differs or improved/enhances “BERT” must also be explained.

·       The answers to the following questions should be given clearly in the manuscript:

o   When a news story changes after it is processed, does the application recognize that? If yes how? If not, what would the implications of this deficiency be?

o   If no article has a picture, a random picture is used. Why is that? Why not leave it as “NULL”?

o   Clearly explain how keyword duplication is prevented. Related with the same topic, the discussion about redundancy wrt to the blends may also be improved.

o   How is blend_embedding  vectorized computed?

o   What are the properties of the data sources used. On the same topic please correct the sentence that first introduces the data sources in section 3.3.1. The sentence asks the reader to recall the data sources employed but the data sources are not listed prior to that.

o   Section 4.3 states that validation of the data elaboration pipeline is done by domain experts. Who? How many? How are they chosen? How reliable is their validation?

·       There is no added value in discussing the ABOUT section of an application. This part can be removed.

·       It could be more engaging if the Discussion section is revised so that it starts by answering the questions what is achieved? What contributions have been made to the field? What sets this work apart and why it is important.

· 

Reviewer 2 Report

Cryptocurrency is every time a more studied topic also from an AI perspective e.g. [1], [2]. In this work the authors propose  Cryptoblend, an AI tool for aggregation and summarization of news on cryptocurrency. The application is publicly available and freely accessible (the URL has been provided). The paper is interesting and well written and just few are the typos to fix (see below).

What it would be interesting is to compare / show the results with those obtained by some of the aggregation and summarization tools of Table 1 (some other tools are mentioned in the related work but not included in the table). Moreover, the quality of the summarization could have been evaluated with one of the standard metrics, e.g. Rouge (in [3] the authors compare 14 automatic evaluation metrics). I suggest the authors to do it in their camera ready version of this paper.

Typos (to be fixed):

line 62: news aggregator+s denote

88: Multi-News ([13]) -> Multi-News [13]

108: Vector AutoRegressive model

Table 1: It exploits Rank algorythm -> It exploits the Rank algorythm

Table 1: Not Applicable ? [16]

226: on CSS template -> on the CSS template 

References on AI-tools on cryptocurrency and summarization (to be possibly cited):

[1] Sawhney R., Agarwal S., Mittal V., Rosso P., Nanda V., Chava S. (2022) Cryptocurrency Bubble Detection: A New Stock Market Dataset, Financial Task & Hyperbolic Models. In: Proc. 2022 Conf. of the North American Chapter of the Association for Computational Linguistics, NAACL-2022, Seattle, Washington, US, July 10-15, pp. 5531-5545

[2] https://pan.webis.de/clef23/pan23-web/author-profiling.html

[3] Fabbri A.R., KryÅ›ciÅ„ski W., McCann B., Xiong C., Socher R., Radev D. (2021). SummEval: Re-evaluating Summarization Evaluation. Transactions of the Association for Computational Linguistics, 9: 391–409

Reviewer 3 Report

This study suggests the creation of a mechanism to compile and synthesize news stories about cryptocurrencies, blockchain technology, and decentralization. This methodology has been used to create a fully functional, open-source web application. To find clusters of related articles, the application uses an agglomerative clustering method along with a BERT-based sentence representation. Additionally, the most representative sentences from each cluster were extracted using a maximum marginal relevance approach. By having professional human readers validate the proposed methodology and application against a set of randomly chosen documents, it has been demonstrated how effective it is.

More details about BERT approach should be added.

The article, in my opinion, is nicely written, with four main chapters that clearly follow the usual style for scientific articles. The material in this study is interesting and straightforward. The delivery is well structured, and the authors have taken care in structuring the material so that readers may simply follow the content. It includes graphics that support and illustrate the background offered here.

There are minor grammar mistakes as:

Line 6: on news - in news

Line 92: is a tool collecting - is a tool for collecting

Line 99: analisys - analysis

Line 193: There seems to be a word order problem here (here discussed - discussed here)

I propose revising references older than five years so that the facts in the post are completely up to date, but other than that, the article is really well written. More related work should be added regarding tools for fraud detections, for example:

- Bratulescu, Razvan-Alexandru, et al. "Fraudulent Activities in the Cyber Realm: DEFRAUDify Project" Proceedings of the 17th International Conference on Availability, Reliability and Security. 2022.

- Barnes, Paul. "Crypto currency and its susceptibility to speculative bubbles, manipulation, scams and fraud." Journal of Advanced Studies in Finance (JASF) 9.2 (2018): 18.

- Sureshbhai, Patel Nikunjkumar, Pronaya Bhattacharya, and Sudeep Tanwar. "KaRuNa: A blockchain-based sentiment analysis framework for fraud cryptocurrency schemes." 2020 IEEE International Conference on Communications Workshops (ICC Workshops). IEEE, 2020.

As future work I would recommend the implementation of more analysis tools that are up-to-date and also make a section on the webpage also to track the impact of news on live price of cryptocurrencies, not only to provide daily news.

Round 2

Reviewer 1 Report

The manuscripts has been thoroughly revised and improved by the authors. The concerns raised are generally dealt with. 

To further improve the manuscript there are three suggestions that can be easily implemented :

1- The language/grammar needs to be checked once more for correcting the residual mistakes (examples from page 11: "The web application here produced...."' "notice that such is skewed...").  This can be achieved if a preferably native speaker other than the authors read the manuscript.

2-Table 1 could be improved by describing all tools in a consistent way (for both short description and data sources used). For example, in description of the tool in row 6, the authors mention the underlying architecture or machine learning paradigm as RNN but for the tool in row 3, there is no info on the architecture, instead the target is mentioned as junk news. It would be preferable if the reader could get the same (or at least similar) information about each tool.

3-Finally the discussion of the architecture in Section 3 should be done with regard to the conceptual diagram given in Figure 1. For instance, either the figure or the discussion may be enhanced to clearly show which part of the diagram is “data elaboration pipeline”. 
